# Fast Approximate Barnes Interpolation:
# Illustrated by Python/Numba Implementation fast-barnes-py v1.0

Bruno K. Zürcher

Federal Office of Meteorology and Climatology MeteoSwiss, Zurich, Switzerland

**Correspondence:** Bruno K. Zürcher (bruno.zuercher@meteoswiss.ch)

**Abstract.** Barnes interpolation is a method that is widely used in geospatial sciences like meteorology to remodel data values recorded at irregularly distributed points into a representative analytical field. When implemented naively, the effort to calculate Barnes interpolation depends on the product of the number of sample points $N$ and the number of grid points $W \times H$, resulting in a computational complexity of $\mathcal{O}(N \cdot W \cdot H)$. In the era of highly resolved grids and overwhelming numbers of sample points, which originate e.g. from the Internet of Things or from crowd-sourced data, this computation can be quite demanding even on high-performance machines.

This paper presents new approaches how very good approximations of Barnes interpolation can be implemented using fast algorithms that have a computational complexity of $\mathcal{O}(N + W \cdot H)$. Two use cases are in particular considered, namely (1) where the used grid is embedded in the Euclidean plane and (2) where the grid is located on the unit sphere.

## 1 Introduction

In the early days of numerical weather prediction, various objective analysis methods were developed to automatically produce the initial conditions from the irregularly spaced observational data (Daley, 1991), one of which was Barnes interpolation (Barnes, 1964). However, objective analysis soon lost its significance in this field against statistical approaches. Today, Barnes method is still used to create grid-based isoline visualizations of geospatial data, like for instance in the meteorological workstation project NinJo (Koppert et al., 2004), which is jointly developed by the Deutscher Wetterdienst, the Bundeswehr Geophysical Service, MeteoSwiss, the Danish Meteorological Institute and the Meteorological Service of Canada.

Barnes interpolation $f(\boldsymbol{x}) : D \longrightarrow \mathbb{R}$ for an arbitrary point $\boldsymbol{x} \in D$ and a given set of sample points $\boldsymbol{x_k} \in D$ with observation values $f_k \in \mathbb{R}$ for $k = 1, ..., N$ is defined as

$$f(\boldsymbol{x}) = \frac{\sum_{k=1}^{N} f_k \cdot w_k(\boldsymbol{x})}{\sum_{k=1}^{N} w_k(\boldsymbol{x})} \tag{1}$$

with Gaussian weights

$$w_k(\boldsymbol{x}) = \mathrm{e}^{-\frac{d(\boldsymbol{x}, \boldsymbol{x_k})^2}{2\sigma^2}}, \tag{2}$$

a distance function $d : D \times D \longrightarrow \mathbb{R}$ and a Gaussian width parameter $\sigma$.

If Barnes interpolation is computed in a straightforward way for a regularly spaced $W \times H$ grid, the computational complexity is given by $\mathcal{O}(N \cdot W \cdot H)$ as easily can be seen from the threefold nested loops of algorithm A given below. Consequently, for big values of $N$ or dense grids, a naive implementation of Barnes interpolation turns out to be unreasonably slow.

---

**Algorithm A** Naive Barnes Interpolation

---

**Input:** sample point coordinates $\boldsymbol{x_k} = (x_k, y_k)$ with sample values $f_k$ for $k = 1, ..., N$ and fall-off parameter $\sigma$.

**Output:** $W \times H$ field $F$ holding Barnes interpolation values.

1: **for** $i = 1$ **to** $W$ **do**

2:      **for** $j = 1$ **to** $H$ **do**

3:          Set numerator $p = 0$ and denominator $q = 0$.

4:          **for** $k = 1$ **to** $N$ **do**

5:              Compute distance between current grid point $\boldsymbol{g}[i, j]$ and current sample point $\boldsymbol{x_k}$ as $d = d(\boldsymbol{g}[i, j], \boldsymbol{x_k})$.

6:              Compute Gaussian weight $w = \mathrm{e}^{-\frac{d^2}{2\sigma^2}}$.

7:              Update $p \mathrel{+}= w \cdot f_k$ and $q \mathrel{+}= w$.

8:          **end for**

9:          Set $F[i, j] = p \, / \, q$.

10:      **end for**

11: **end for**

12: **return** $F$.

---

The fact that the Gaussian weight function quickly approaches $0$ for increasing distances leads to a first improvement attempt, which consists in neglecting all terms in the sums of (1), for which the weights $w_k$ drop below a certain limit $w_0$, e.g. $w_0 = 0.001$. This is equivalent to take only observation points $\boldsymbol{x_k}$ into account that lie within the distance $r_0 = \sigma\sqrt{-2\ln w_0}$ from the interpolation point $\boldsymbol{x}$. Thus, the described procedure requires the ability to quickly extract all observation points $\boldsymbol{x_k}$ that lie within a distance $r_0$ from point $\boldsymbol{x}$. Data structures that support such searches are e.g. so called k-d-trees (Bentley, 1975) or quadtrees (Finkel and Bentley, 1974).

This improved approach actually reduces the required computation time by a constant factor, but the computational complexity remains in the same order. To see this, note that a specific sample point $\boldsymbol{x_k}$ contributes to the interpolation value of exactly those grid points that are contained in the circular disk $B_{r_0}(\boldsymbol{x_k}) = \{\boldsymbol{q} \in D \mid d(\boldsymbol{x_k}, \boldsymbol{q}) < r_0\}$ of radius $r_0$ around it. Note also that in a regularly spaced grid the number of affected grid points is roughly the same for each sample point. If now the number of grid points $W$ and $H$ is in each dimension increased by a factor $\kappa$ – i.e. the grid becomes denser – the number of grid points contained in $B_{r_0}(\boldsymbol{x_k})$ grows accordingly, namely by a factor $\kappa^2$, which shows that the algorithmic costs rise in direct dependency to $W$ and $H$. Since this is obviously as well true for the number of sample points $N$, the improved algorithm also has complexity $\mathcal{O}(N \cdot W \cdot H)$.

In this paper, we discuss a new and fast technique to compute very good approximations of Barnes interpolation. The utilized underlying principle of applying multiple convolution passes with a simple rectangular filter in order to approximate

a Gaussian convolution is well known in computational engineering. In image processing and computer vision, for instance, Gaussian filtering of images is often efficiently calculated by repeated application of an averaging filter (Wells, 1986).

The theoretical background of the new approach is presented in Sect. 2 and 3. After, we investigate two use cases for the domain $D$,

(i) $D = \mathbb{R}^2$ the Euclidean plane with the usual distance $d(\boldsymbol{p}, \boldsymbol{q}) = \|\boldsymbol{p} - \boldsymbol{q}\|_2$ in full detail in Sect. 4 and 5.4,

(ii) $D = \mathcal{S}^2 = \{\boldsymbol{p} \in \mathbb{R}^3 \mid \|\boldsymbol{p}\|_2 = 1\}$ the unit sphere with $d(\boldsymbol{p}, \boldsymbol{q})$ the spherical distance between $\boldsymbol{p}$ and $\boldsymbol{q}$ as a broad outline in Sect. 5.5.

## 2  Conclusions from Central Limit Theorem

For a set $\{X_k\}_{k=1}^n$ of independent and identically distributed random variables with mean $\mu$ and variance $\sigma^2$, the central limit theorem (Klenke, 2020) states that the probability distribution of their sum converges to a normal distribution, if $n$ approaches infinity, formally

$$P\left[\frac{X_1 + \cdots + X_n}{\sqrt{n}} \leq a\right]$$

$$\xrightarrow[n \to \infty]{} \frac{1}{\sqrt{2\pi}\sigma} \int_{-\infty}^{a} \mathrm{e}^{-\frac{1}{2}\left(\frac{t-\mu}{\sigma}\right)^2} dt. \tag{3}$$

Without loss of generality we assume in the further discussion $\mu = 0$. Let $p(x)$ denote the PDF (probability density function) of the scaled random variables $\{\frac{1}{\sqrt{n}} X_k\}_{k=1}^n$ that consequently have the variance $\frac{\sigma^2}{n}$. Since the PDF of a sum of random variables corresponds to the convolution of their individual PDFs, we find on the other hand

$$P\left[\frac{X_1 + \cdots + X_n}{\sqrt{n}} \leq a\right] = \int_{-\infty}^{a} p^{*n}(x)\, dx,$$

where $p^{*n}(x)$ denotes the $n$-fold convolution of $p(x)$ with itself (refer to appendix A). With that result we can write relationship (3) equivalently in an unintegrated form as

$$p^{*n}(x) \xrightarrow[n \to \infty]{} \frac{1}{\sqrt{2\pi}\sigma} \mathrm{e}^{-\frac{x^2}{2\sigma^2}}, \tag{4}$$

which leads directly to

**Approximation 1.** *For sufficiently large $n$, the $n$-fold self-convolution of a probability density function $p(x)$ with mean $\mu = 0$*
*and variance $\frac{\sigma^2}{n}$ approximates a Gaussian with mean $0$ and variance $\sigma^2$, i.e.*

$$p^{*n}(x) \approx \frac{1}{\sqrt{2\pi}\sigma} \mathrm{e}^{-\frac{x^2}{2\sigma^2}}.$$

Note that this approximation is valid for arbitrary PDFs $p(x)$ with mean $\mu = 0$ and variance $\frac{\sigma^2}{n}$.

A particular simple PDF is given by the uniform distribution. We therefore define a family of uniform PDFs $\{u_n(x)\}_{n=1}^{\infty}$, of which each member $u_n(x)$ has mean 0 and variance $\frac{\sigma^2}{n}$. These uniform PDFs can be expressed by means of elementary rectangular functions

$$r_n(x) = \begin{cases} 1 & \text{for } |x| \leq \sqrt{\frac{3}{n}}\,\sigma \\ 0 & \text{otherwise} \end{cases} \quad \text{where } n = 1, 2, \cdots , \tag{5}$$

such that

$$u_n(x) = \frac{1}{2\sqrt{\frac{3}{n}}\,\sigma} r_n(x) = \begin{cases} \frac{1}{2\sqrt{\frac{3}{n}}\,\sigma} & \text{for } |x| \leq \sqrt{\frac{3}{n}}\,\sigma \\ 0 & \text{otherwise} \end{cases}$$

where $n = 1, 2, \cdots$. From this definition it is clear that $u_n(x)$ is actually a PDF with mean $\mathrm{E}[u_n] = 0$ and variance

$$\mathrm{Var}(u_n) = \int_{-\infty}^{\infty} x^2 \cdot u_n(x)\, dx = \frac{1}{2\sqrt{\frac{3}{n}}\,\sigma} \int_{-\sqrt{\frac{3}{n}}\,\sigma}^{\sqrt{\frac{3}{n}}\,\sigma} x^2 \, dx$$

$$= \frac{1}{2\sqrt{\frac{3}{n}}\,\sigma} \cdot \frac{1}{3} x^3 \Big|_{-\sqrt{\frac{3}{n}}\,\sigma}^{\sqrt{\frac{3}{n}}\,\sigma} = \frac{1}{2\sqrt{\frac{3}{n}}\,\sigma} \cdot \frac{2}{3} \left( \sqrt{\frac{3}{n}}\,\sigma \right)^3 = \frac{\sigma^2}{n}$$

as postulated. According to convergence relation (4) and approximation 1, the series of the $n$-fold self-convolutions $\{u_n^{*n}(x)\}_{n=1}^{\infty}$ converges to a Gaussian with mean 0 and variance $\sigma^2$. The converging behavior can actually be examined visually in Fig. 1, which plots the $n$-fold self-convolution of the first few family members.

The central limit theorem can also be stated more generally for i.i.d. $m$-dimensional random vectors $\{\boldsymbol{X_k}\}_{k=1}^n$, refer for instance to (Muirhead, 1982; Klenke, 2020). Supposing the $\boldsymbol{X_k}$ to have a mean vector $\boldsymbol{\mu} = \boldsymbol{0}$ and a covariance matrix $\boldsymbol{\Sigma}$, we can follow the same line of argument as in the one-dimensional case. Let $p(\boldsymbol{x})$ be the joint PDF of the scaled random variables $\{\frac{1}{\sqrt{n}}\boldsymbol{X_k}\}_{k=1}^n$, which therefore have a zero mean vector and the covariance matrix $\frac{1}{n}\boldsymbol{\Sigma}$. Then the limit law becomes in $m$ dimensions

$$p^{*n}(\boldsymbol{x}) \xrightarrow[n \to \infty]{} \frac{1}{(2\pi)^{\frac{m}{2}} \sqrt{\det \boldsymbol{\Sigma}}} \, \mathrm{e}^{-\frac{1}{2}\boldsymbol{x}^T \boldsymbol{\Sigma}^{-1} \boldsymbol{x}}.$$

For the remainder of the discussion, we fix the number of dimensions to $m = 2$ and for the sake of readability, we write the vector argument $\boldsymbol{x}$ in its component form $(x, y)$, if it is appropriate. In case the random vectors $\boldsymbol{X_k}$ are isotropic, i.e. do not have any preference in a specific spatial direction, the covariance matrix is a multiple of the identity matrix $\boldsymbol{\Sigma} = \sigma^2 \boldsymbol{I}$ and the limit law simplifies in two dimensions to

$$p^{*n}(\boldsymbol{x}) \xrightarrow[n \to \infty]{} \frac{1}{2\pi\sigma^2} \, \mathrm{e}^{-\frac{1}{2\sigma^2} \|\boldsymbol{x}\|^2}. \tag{6}$$

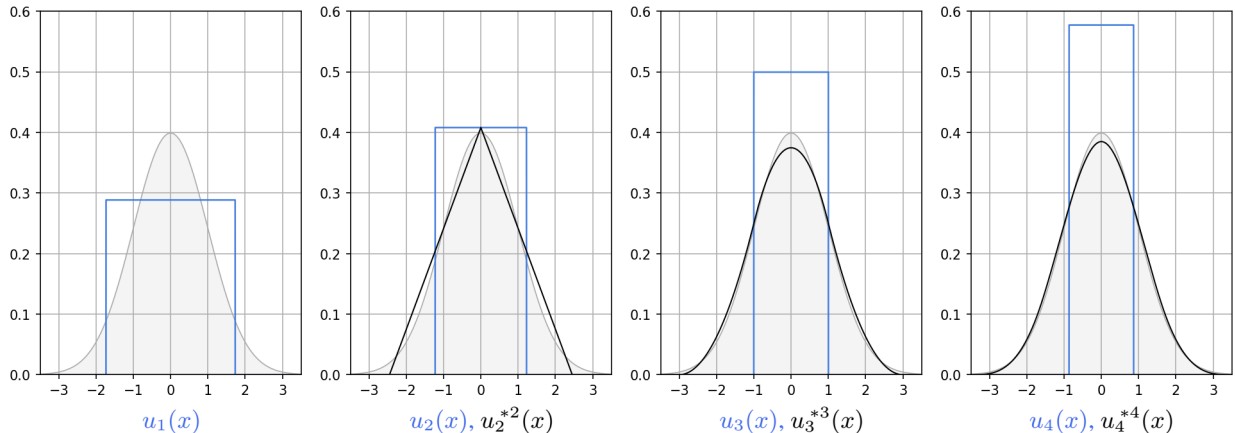

**Figure 1.** From left to right: In blue the plot of the PDFs of the uniform distributions $u_1(x)$, $u_2(x)$, $u_3(x)$ and $u_4(x)$ for $\sigma = 1$. In black their self-convolutions $u_2^{*2}(x)$, $u_3^{*3}(x)$ and $u_4^{*4}(x)$. The area covered by the PDF of the normal distribution is indicated in grey.

In the following step, we are aiming to substitute $p(\boldsymbol{x})$ with the members of the family $\{u_n^{(2)}(\boldsymbol{x})\}_{n=1}^{\infty}$ of two-dimensional uniform distributions over a square-shaped domain, which are defined as

$$95 \quad u_n^{(2)}(x,y) = u_n(x) \cdot u_n(y) = \frac{n}{12\,\sigma^2}\, r_n(x) \cdot r_n(y)$$

$$= \begin{cases} \frac{n}{12\sigma^2} & \text{for } |x|,|y| \leq \sqrt{\frac{3}{n}}\,\sigma \\ 0 & \text{otherwise} \end{cases} \quad \text{for } n = 1, 2, \cdots.$$

With this definition, the members $u_n^{(2)}$ have a mean vector $\boldsymbol{0}$ and an isotropic covariance matrix $\frac{\sigma^2}{n}\boldsymbol{I}$ and hence satisfy the prerequisite of limit law (6). Note also that $u_n^{(2)}(\boldsymbol{x})$ is separable because $u_n^{(2)}(x,y) = u_n(x) \cdot u_n(y)$. As a consequence of the latter, the $n$-fold self-convolution of $u_n^{(2)}(\boldsymbol{x})$ is itself separable, i.e.

$$100 \quad \left(u_n^{(2)}\right)^{*n}(x,y) = \left(u_n(x) \cdot u_n(y)\right)^{*n}$$

$$= u_n^{\overset{x}{*}n}(x) \cdot u_n^{\overset{y}{*}n}(y)$$

$$= \left(\frac{n}{12\,\sigma^2}\right)^n r_n^{\overset{x}{*}n}(x) \cdot r_n^{\overset{y}{*}n}(y), \tag{7}$$

where the operators $\overset{x}{*}$ denote one-dimensional convolution in x-direction and $\overset{y}{*}$ one-dimensional convolution in y-direction, respectively (refer to appendix B). Substituting $p(\boldsymbol{x})$ in (6) with the r.h.s. of (7) we obtain

$$105 \quad \left(\frac{n}{12\,\sigma^2}\right)^n r_n^{\overset{x}{*}n}(x) \cdot r_n^{\overset{y}{*}n}(y) \quad \xrightarrow[n \to \infty]{} \quad \frac{1}{2\pi\sigma^2}\, \mathrm{e}^{-\frac{1}{2\sigma^2}\|\boldsymbol{x}\|^2},$$

or expressed as

**Approximation 2.** *For sufficiently large $n$, the $n$-fold self-convolution of the two-dimensional uniform probability density function $\frac{n}{12\sigma^2} r_n(x) \cdot r_n(y)$ approximates a bivariate Gaussian with mean vector $\mathbf{0}$ and covariance matrix $\sigma^2 \mathbf{I}$, i.e.*

$$\left(\frac{n}{12\sigma^2}\right)^n r_n^{\overset{x}{*}n}(x) \cdot r_n^{\overset{y}{*}n}(y) \approx \frac{1}{2\pi\sigma^2}\, e^{-\frac{1}{2\sigma^2}\|\boldsymbol{x}\|^2},$$

*where $r_n(x)$ is an elementary rectangular function defined as*

$$r_n(x) = \begin{cases} 1 & for\ |x| \leq \sqrt{\frac{3}{n}}\,\sigma \\ 0 & otherwise \end{cases} \quad for\ n = 1, 2, \cdots .$$

## 3  Barnes Interpolation as Series of Convolutions

Let $\varphi_{\boldsymbol{\mu},\sigma}(\boldsymbol{x})$ denote the PDF of a two-dimensional normal distribution with mean vector $\boldsymbol{\mu}$ and isotropic variance $\sigma^2$, i.e.

$$\varphi_{\boldsymbol{\mu},\sigma}(\boldsymbol{x}) = \frac{1}{2\pi\sigma^2}\, e^{-\frac{1}{2\sigma^2}\|\boldsymbol{x}-\boldsymbol{\mu}\|^2}.$$

Note that $\varphi_{\mathbf{0},\sigma}(\boldsymbol{x})$ corresponds to the r.h.s. of approximation 2. Further let $\delta_{\boldsymbol{a}}(\boldsymbol{x})$ denote the Dirac or unit impulse function at location $\boldsymbol{a}$ with the property $\delta_{\boldsymbol{a}} * f(\boldsymbol{x}) = f(\boldsymbol{x} - \boldsymbol{a})$. Then we can write

$$\varphi_{\boldsymbol{\mu},\sigma}(\boldsymbol{x}) = \frac{1}{2\pi\sigma^2}\, e^{-\frac{1}{2\sigma^2}\|\boldsymbol{x}-\boldsymbol{\mu}\|^2} = \delta_{\boldsymbol{\mu}} * \left(\frac{1}{2\pi\sigma^2}\, e^{-\frac{1}{2\sigma^2}\|\boldsymbol{x}\|^2}\right)$$

$$= \delta_{\boldsymbol{\mu}} * \varphi_{\mathbf{0},\sigma}(\boldsymbol{x}).$$

Thus, a Gaussian weighted sum as found in the numerator of Barnes interpolation (1) for the Euclidean plane $\mathbb{R}^2$ can be written as convolutional operation

$$\sum_{k=1}^{N} f_k \cdot e^{-\frac{1}{2\sigma^2}\|\boldsymbol{x}-\boldsymbol{x_k}\|^2} = 2\pi\sigma^2 \sum_{k=1}^{N} f_k \cdot \varphi_{\boldsymbol{x_k},\sigma}(\boldsymbol{x})$$

$$= 2\pi\sigma^2 \sum_{k=1}^{N} f_k \cdot (\delta_{\boldsymbol{x_k}} * \varphi_{\mathbf{0},\sigma})(\boldsymbol{x})$$

and due to the distributivity and the associativity with scalar multiplication of the convolution operator follows

$$= 2\pi\sigma^2 \sum_{k=1}^{N} (f_k \cdot \delta_{\boldsymbol{x_k}}) * \varphi_{\mathbf{0},\sigma}(\boldsymbol{x})$$

$$= 2\pi\sigma^2 \left(\sum_{k=1}^{N} f_k \cdot \delta_{\boldsymbol{x_k}}\right) * \varphi_{\mathbf{0},\sigma}(\boldsymbol{x}). \tag{8}$$

Substituting $\varphi_{\mathbf{0},\sigma}(\boldsymbol{x})$ with approximation 2, we obtain for sufficiently large $n$

$$\sum_{k=1}^{N} f_k \cdot e^{-\frac{1}{2\sigma^2}\|\boldsymbol{x}-\boldsymbol{x_k}\|^2}$$

$$\approx 2\pi\sigma^2 \left(\frac{n}{12\sigma^2}\right)^n \left(\sum_{k=1}^{N} f_k \cdot \delta_{\boldsymbol{x_k}}\right) * \left(r_n^{\overset{x}{*}n}(x) \cdot r_n^{\overset{y}{*}n}(y)\right).$$

For the denominator of Barnes interpolation (1) we can use the same expression, but set the coefficients $f_k$ to 1. Since the common factors in the numerator and the denominator cancel each other, we can state

**Approximation 3.** *For sufficiently large $n$, Barnes interpolation for the Euclidean plane $\mathbb{R}^2$ can be approximated by*

$$f(x,y) \approx \frac{\left(\sum_{k=1}^{N} f_k \cdot \delta_{\boldsymbol{x_k}}\right) * \left(r_n \overset{x}{*}{}^n(x) \cdot r_n \overset{y}{*}{}^n(y)\right)}{\left(\sum_{k=1}^{N} \delta_{\boldsymbol{x_k}}\right) * \left(r_n \overset{x}{*}{}^n(x) \cdot r_n \overset{y}{*}{}^n(y)\right)}, \tag{9}$$

*provided that the quotient is defined.*

In other words, Barnes interpolation can very easily be approximated by the quotient of two convolutional expressions, both consisting of an irregularly spaced Dirac-comb, followed by a sequence of convolutions with a one-dimensional rectangular function of width $2\sigma\sqrt{3/n}$, executed $n$-times in x-direction and $n$-times in y-direction. As the convolution operation is commutative, the convolutions can basically be carried out in any order. The sequence shown in approximation 3, evaluated from left to right, is however especially favorable regarding the computational effort.

Approximation 3 can as well be stated in a more generalized context, i.e. also for non-uniform PDFs. In the special case of using a normal distribution with mean 0 and variance $\sigma^2$, we can even formulate a convolutional expression that is equal to Barnes interpolation. Refer to appendix C for more details.

## 4 Discretization

Approximation 3 leads in a straightforward way to a very efficient algorithm for an approximate computation of Barnes interpolation on a regular grid $\Gamma$ that is embedded in the Euclidean plane $\mathbb{R}^2$. Let

$$\Gamma = \left\{ (i \cdot \Delta s, j \cdot \Delta s) \in \mathbb{R}^2 \mid 0 \leq i < W, 0 \leq j < H \right\},$$

be a grid of dimension $W \times H$ with a grid point spacing $\Delta s$. Without loss of generality, we assume that all sample points $\boldsymbol{x_k}$ are sufficiently good contained in the interior of $\Gamma$. In what follows, we differentiate discrete functions from their continuous function counterparts by enclosing the arguments in brackets instead of parentheses and write, for instance, $g[i]$ for $g(x)$ in the one-dimensional case and $g[i,j]$ for $g(x,y)$ in the two-dimensional case, respectively.

We now compute in an iterative procedure for each point $[i,j]$ of grid $\Gamma$ the convolutional expression that corresponds to the numerator (or denominator) of approximation 3. The intermediate fields that result from this iteration are denoted by $F^{(m)}$, where $m$ indicates the iteration stage.

The first step consists in discretizing the expression $\sum f_k \cdot \delta_{\boldsymbol{x_k}}$, i.e. in injecting the values $f_k$ at their respective sample points $\boldsymbol{x_k}$ into the underlying grid. For this purpose, the field $F^{(0)}$ is initialized with 0. From the definition of Dirac's impulse function

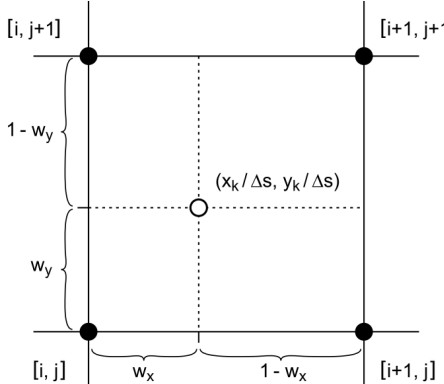

**Figure 2.** The nearer a sample point $x_k$ is located to a grid point, the larger the weight assigned to it. Grid point $[i, j]$ e.g. receives a weight of $(1 - w_x) \cdot (1 - w_y)$.

in two dimensions

$$\delta_{\mathbf{0}}(x, y) = \lim_{\alpha \to 0} \frac{1}{\alpha^2} r_\alpha(x, y)$$

$$\text{where } r_\alpha(x, y) = \begin{cases} 1 & \text{for } |x|, |y| \leq \frac{1}{2\alpha} \\ 0 & \text{otherwise} \end{cases},$$

we deduce for its discrete version that the grid cell containing the considered sample point receives the weight $1/\Delta s^2$, while all other cells are left unchanged with weight $0$. Since a sample point $x_k$ in general does not coincide with a grid point (refer

also to Fig. 2) and in order to achieve a good localization, the Dirac impulse $\delta_{x_k}$ is distributed in a bilinear way to its four neighbouring grid points according to step 5 of algorithm B.1.

Note that if a grid point $[i, j]$ is affected by several sample points, the determined weight fractions are accumulated accordingly in the respective field element $F^{(0)}[i, j]$. For the final calculation of quotient (9), the factor $1/\Delta s^2$ cancels out and can therefore be omitted in algorithm B.1, but for reasons of mathematical correctness it is shown here. Since we have $N$ input

points and we perform a fixed number of operations for each of them, the complexity of algorithm B.1 is given by $\mathcal{O}(N)$.

The other algorithmic fragment we require, which is implemented by algorithm B.2, is the computation of a one-dimensional convolution of an arbitrary function $g(x)$ with the rectangular function $r_n(x)$ as defined in (5). Employing the definition of convolution, we obtain

$$h(x) = g * r_n(x) = \int_{-\infty}^{\infty} g(x - t) \cdot r_n(t) dt$$

$$= \int_{-\tau}^{\tau} g(x - t) dt = \int_{x-\tau}^{x+\tau} g(t) dt \tag{10}$$

**Algorithm B.1** Inject observation values $f_k$ into grid $\Gamma$

---

**Input:** sample point coordinates $\boldsymbol{x_k} = (x_k, y_k)$ with sample values $f_k$ for $k = 1, ..., N$.

**Output:** $W \times H$ field $F^{(0)}$ with injected sample values.

1: Initialize field $F^{(0)}$ with 0.

2: **for** $k = 1$ **to** $N$ **do**

3:     Determine indices $i = \lfloor x_k / \Delta s \rfloor$ and $j = \lfloor y_k / \Delta s \rfloor$ of lower left neighbouring grid point.

4:     Compute weights $w_x = x_k / \Delta s - i$ and $w_y = y_k / \Delta s - j$, which are both contained in $[0, 1)$.

5:     Distribute sample value $f_k$ in bilinear way among four neighbouring grid points, i.e.

$$F^{(0)}[i, j] \mathrel{+}= (1 - w_x) \cdot (1 - w_y) \cdot f_k / \Delta s^2$$
$$F^{(0)}[i, j + 1] \mathrel{+}= (1 - w_x) \cdot w_y \cdot f_k / \Delta s^2$$
$$F^{(0)}[i + 1, j] \mathrel{+}= w_x \cdot (1 - w_y) \cdot f_k / \Delta s^2$$
$$F^{(0)}[i + 1, j + 1] \mathrel{+}= w_x \cdot w_y \cdot f_k / \Delta s^2$$

6: **end for**

7: **return** $F^{(0)}$.

---

where $\tau = \sigma \sqrt{3/n}$. With other words, the convolution $g * r_n$ at point $x$ is simply the integral of $g(x)$ in the window $[x - \tau, x + \tau]$. Transferred to a one-dimensional grid with spacing $\Delta s$, the rectangular function $r_n(x)$ reads as rectangular pulse

$$r_T[k] = \begin{cases} 1 & \text{for } |k| \leq T \\ 0 & \text{otherwise} \end{cases} \quad k \in \mathbb{Z},$$

with a width parameter $T \in \mathbb{N}_0$ that is gained by rounding $\tau / \Delta s$ to the nearest integer number

$$T = \left\lfloor \frac{\tau}{\Delta s} + \frac{1}{2} \right\rfloor = \left\lfloor \sqrt{\frac{3}{n}} \frac{\sigma}{\Delta s} + \frac{1}{2} \right\rfloor. \tag{11}$$

Then equation (10) translates in the discrete case to

$$h[k] = g * r_T[k] = \sum_{i=-\infty}^{\infty} g[k - i] \cdot r_T[i] \cdot \Delta s$$
$$= \sum_{i=-T}^{T} g[k - i] \cdot \Delta s = \sum_{i=k-T}^{k+T} g[i] \cdot \Delta s,$$

where element $h[k]$ corresponds to $h(k \cdot \Delta s)$ and $g[i]$ to $g(i \cdot \Delta s)$. Equivalently to the continuous case, the value $h[k]$ results up to a factor $\Delta s$ from putting a window of length $2T + 1$ centrally over the sequence element $g[k]$ and summing up all elements covered by it.

Assuming that we already computed $h[k - 1]$, it is immediately clear that the following value $h[k]$ results from moving the window by one position to the right and thus can be obtained very easily from $h[k - 1]$ by adding the newly enclosed sequence element $g[k + T]$, but subtracting element $g[k - T - 1]$ that falls outside the window.

**Algorithm B.2** Convolution of a sequence $g[k]$ with rectangular pulse of length $2T+1$

**Input:** sequence $g[k]$ with $k = 1, ..., L$ and length $2T+1$ of rectangular pulse.

**Output:** the convolution $g * r_T$.

1: Compute $w = \sum_{i=-T}^{T} g[1-i]$, where all elements $g[k]$ which are not defined are set to 0.

2: Set $h[1] = w \cdot \Delta s$.

3: **for** $k = 2$ **to** $L$ **do**

4:      Update $w += g[k+T] - g[k-T-1]$, where all elements $g[k]$ which are not defined are set to 0.

5:      Set $h[k] = w \cdot \Delta s$.

6: **end for**

7: **return** sequence $h[k]$ with $k = 1, ..., L$, which is the convolution $g * r_T$.

As in the case of algorithm B.1 before, the factor $\Delta s$ cancels in the final calculation of (9) and can therefore also be omitted here. Thus, algorithm B.2 has $2T$ additions in step 1 and another $2(L-1)$ additions in the loop of step 3. Assuming that $T$ is much smaller than $L$, an algorithmic complexity of $\mathcal{O}(L)$ results.

Now we are able to formulate algorithm B.3 that computes convolutional expressions as found in the numerator and denominator of approximation 3.

**Algorithm B.3** $n$-fold convolution of a field $F^{(0)}[i,j]$ with a two-dimensional rectangular pulse

**Input:** $W \times H$ input field $F^{(0)}$, the length of the rectangular pulse $2T+1$ and the number of convolutions $n$ to be carried out.

**Output:** $n$-fold convolved $W \times H$ field $F^{(n)}$, which is equal to $F^{(0)} * (r_T^{\overset{x}{*}n} \cdot r_T^{\overset{y}{*}n})$.

1: **for** $k = 1$ **to** $n$ **do**

2:      Rename the $(k-1)$-fold convolved field $F^{(k-1)}$ to $F$.

3:      **for** $i = 1$ **to** $W$ **do**

4:          Convolve $i$-th field row $F[i,.]$ according to algorithm B.2 with rectangular pulse $r_T$ and store result back in respective field row.

5:      **end for**

6:      **for** $j = 1$ **to** $H$ **do**

7:          Convolve $j$-th field column $F[.,j]$ according to algorithm B.2 with rectangular pulse $r_T$ and store result back in respective field column.

8:      **end for**

9:      Rename $F$, which is now the $k$-fold convolved field, to $F^{(k)}$.

10: **end for**

11: **return** $F^{(n)}$.

Note that due to the commutativity of $\overset{x}{*}$ and $\overset{y}{*}$, the outer loop over index $k$ can be moved inward within the loops over the rows and the columns, respectively. With this alternate loop layout, the field is first traversed row-wise in a single pass, where each row is in one sweep $n$-times convolved with $r_T$. Subsequently, the field is traversed column-wise and each column is

195 $n$-times convolved. In such a way, more economic strategies with respect to memory access can be achieved and moreover, this loop order is very well suited for parallel execution, such that algorithm B.3 can be computed very efficiently. Since in practice $n$ is chosen constant (proven values for $n$ lie between 3 and 6), the algorithmic complexity is $\mathcal{O}(W \cdot H)$.

Algorithms B.1 and B.3 now allow us to state the final algorithm B, which implements the approximate computation of 200 Barnes interpolation (9).

---

**Algorithm B** Approximation of Barnes interpolation

**Input:** sample point coordinates $\boldsymbol{x_k} = (x_k, y_k)$ with sample values $f_k$ for $k = 1, ..., N$, the number of iterations $n$ and fall-off parameter $\sigma$.

**Output:** $W \times H$ field $F$ that approximates Barnes interpolation.

1: Set $T = \left\lfloor \sqrt{\frac{3}{n}} \frac{\sigma}{\Delta s} + \frac{1}{2} \right\rfloor$.

2: To obtain $n$-fold convolved numerator field $P^{(n)}$ do:

3:     Determine initial numerator field $P^{(0)}$ by invoking algorithm B.1 and injecting sample values $\{f_k\}_{k=1}^{N}$.

4:     Compute $P^{(n)}$ by invoking algorithm B.3 with field $P^{(0)}$ and the rectangular pulse width $2T + 1$.

5: To obtain $n$-fold convolved denominator field $Q^{(n)}$ do:

6:     Determine initial denominator field $Q^{(0)}$ by invoking algorithm B.1 and injecting constant sample values $\{1\}_{k=1}^{N}$.

7:     Compute $Q^{(n)}$ by invoking algorithm B.3 with field $Q^{(0)}$ and the rectangular pulse width $2T + 1$.

8: Compute field $F$ by dividing $P^{(n)}$ and $Q^{(n)}$ element-wise, i.e. by setting $F[i,j] = P^{(n)}[i,j] / Q^{(n)}[i,j]$.

9: **return** $F$.

---

If the denominator $Q^{(n)}[i,j]$ in step 8 is 0, which is the case if the grid point $[i,j]$ has at least in one dimension a greater distance than $2nT$ from the nearest sample point, the corresponding field value $F[i,j]$ is undefined and set to NaN.

Since algorithm B.1 and B.3 are invoked twice and step 8 employs another $W \cdot H$ divisions, the overall algorithmic complexity of the presented approach is limited to $\mathcal{O}(N + W \cdot H)$, which is a drastic improvement compared to the costs of $\mathcal{O}(N \cdot W \cdot H)$ 205 of the naive implementation.

## 5 Results and Further Considerations

### 5.1 Test Setup

The described algorithm B – hereinafter denoted with "convolution" – was tested on a dataset that contained in total 3490 QFF values (air pressure reduced to mean sea level) obtained from measurements at meteorological stations distributed over Europe 210 and dating from the 27 July 2020 at 12:00 UTC. More specifically, we considered the geographic area $D = [-26°E, 49°E] \times$

$[34.5°N, 72°N] \subset \mathbb{R}^2$ equipped with the Euclidean distance function defined on $D \times D$, i.e. we measured distances in a first examination directly in the plate carrée projection neglecting the curved shape of the earth. The values of the QFF data range from 992.1 hPa to 1023.2 hPa.

The convolution interpolation algorithm is subsequently compared with the results of two alternate algorithms. The first of them – referred as "naive" – is given by the naive algorithm A as stated in the introduction. The second one – denoted with "radius" – consists of the improved naive algorithm that considers in its innermost loop only those observation points whose Gaussian weights $w$ exceed $0.001$ and thus are located within a radius of $3.717\,\sigma$ around the interpolation point. For this purpose, the implementation performs a so-called radius search on a k-d-tree, which contains all observation points. Such a radius search can be achieved with a worst case complexity of $\mathcal{O}(\sqrt{N})$.

All algorithms were implemented in Python using the Numba just-in-time compiler (Lam et al., 2015) in order to achieve compiled-code performance using ordinary Python code. The tests were conducted on a computer with a customary 2.6 GHz Intel i7-6600U processor with two cores, which is in fact only of minor importance since the tested code was written in single-threaded manner. All time measurements were performed ten times and the best value among them was set as the final execution time of the respective algorithm.

## 5.2 Visual Results

In general, Barnes method yields a remarkable good interpolation and results in an aesthetic illustration for regions where the distance between the sample points has the same order of magnitude as the used Gaussian width parameter $\sigma$. However, if the distance between adjacent sample points is large compared to $\sigma$, this method exhibits some shortcomings because then the interpolation converges towards a step function with steep transitions. This effect can be clearly identified, for example, in the generation of plateaus of almost constant value over the Atlantic ocean in Fig. 3a. In the limit case, if $\sigma \to 0$, the interpolation produces a Voronoi tessellation with cells of constant value around a sample point that are bordered by discontinuities towards the neighbouring cells.

The comparison of the isoline visualizations in Figs. 3a and 3b shows in the well-defined areas an excellent agreement between the two approaches. The result for the radius algorithm is similarly consistent with the other two and is therefore not depicted.

Note that the shaded, i.e. the undefined areas of Fig. 3b correspond to those areas, where Barnes interpolation produces the plateau effect. In this sense, one can state that the convolution algorithm filters out the problematic areas in an inherent way.

## 5.3 Time Measurements

For a grid of constant size, the measured execution times in Table 1 show for the naive and the radius algorithm a linear dependence to the number $N$ of considered sample points, while they are almost constant for the presented convolution algorithm. The costs of the injection step are obviously more or less inexistent compared to the costs of the subsequent steps of the algorithm. This fact is in entire agreement to the deduced complexity $\mathcal{O}(N + W \cdot H)$, since in our test setup the grid size $W \times H$ clearly dominates over $N$.

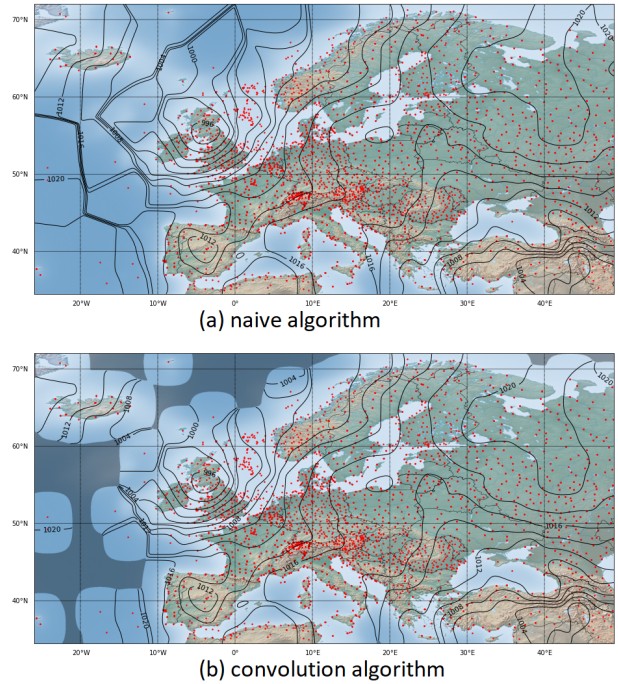

(a) naive algorithm

(b) convolution algorithm

**Figure 3.** Image (a) shows exact Barnes interpolation with naive algorithm for 3490 sample points depicted in red, a $2400 \times 1200$ grid with a resolution 32 grid points$/°$ and $\sigma = 1.0°$.

Image (b) shows approximate Barnes interpolation with convolution algorithm for the same settings as for the naive algorithm above. The applied 4-fold convolution uses a rectangle mask of size 57. Areas where denominator of (9) drops below a value of 0.001 or is even 0 are rendered with a darker shade.

| Number of | Algorithm | | |
|---|---|---|---|
| **Sample Points** | Naive | Radius | Convol. |
| 54 | 6.198 | 0.961 | 0.247 |
| 218 | 21.558 | 1.776 | 0.248 |
| 872 | 78.407 | 4.097 | 0.245 |
| 3490 | 280.764 | 11.840 | 0.247 |

**Table 1.** Execution times (in $s$) of the investigated algorithms for varying numbers of sample points. The grid size of $2400 \times 1200$ points with a resolution of 32 points$/°$ and the Gaussian width $\sigma = 1.0°$ are kept constant. The convolution algorithm applied a 4-fold convolution.

Note also that the speed-up factor between the naive and the convolution algorithm ranges for the considered number of
245 sample points roughly between 25 and 1000.

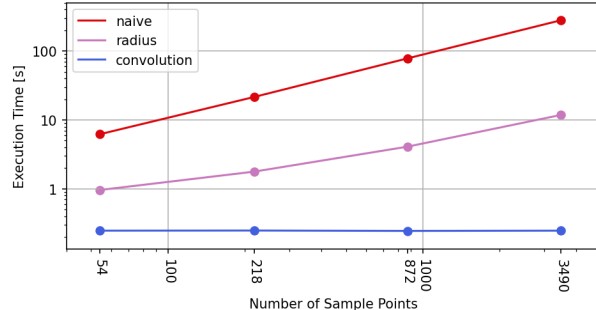

**Figure 4.** Plot of execution times from Table 1 against number of sample points. Both axis use a logarithmic scale.

If the grid size is varied, all considered algorithms reveal a linear-like dependence to the number of points in the grid as can be seen in Table 2 and Fig. 5. For smaller grid sizes, the convolution algorithm provides a speed-up factor of around 2000 compared to the naive implementation, but for bigger grids the factor drops below 1000.

This effect can be explained by the fact that the crucial parts of the convolution algorithm access memory for smaller grids with a high spatial and temporal locality and thus making optimal use of the highly efficient CPU cache memory (Patterson and Hennessy, 2014). For bigger grids the number of cache misses increases, which result in a slightly degraded performance.

| Grid Size | Resol. | Algorithm | | |
|---|---|---|---|---|
| | | Naive | Radius | Convol. |
| $300 \times 150$ | 4 pt/° | 4.415 | 0.203 | 0.002 |
| $600 \times 300$ | 8 pt/° | 17.626 | 0.782 | 0.011 |
| $1200 \times 600$ | 16 pt/° | 70.871 | 3.031 | 0.047 |
| $2400 \times 1200$ | 32 pt/° | 283.735 | 11.881 | 0.247 |
| $4800 \times 2400$ | 64 pt/° | 1134.265 | 47.044 | 1.261 |

**Table 2.** Execution times (in $s$) of the investigated algorithms for varying grid sizes and resolutions, respectively. The number of sample points $N = 3490$ and the Gaussian width $\sigma = 1.0°$ are kept constant. The convolution algorithm applied a 4-fold convolution.

As to be expected, the Gaussian width parameter $\sigma$ has no decisive impact on the execution times measured for the naive and

the convolution algorithm (refer to Table 3 and Fig. 6). The radius algorithm, on the other hand, shows a quadratic dependence, since the relevant area around a grid point – and thus also the average number of sample points to be considered – grows quadratically with the radius of influence, which in turn depends linearly on $\sigma$.

The unstable time measurements for the naive algorithm are caused by a peculiar execution time behavior of the exponential function. Although for distances $d > 38.6\sigma$, the value of (2) is less than the smallest representable float and therefore results

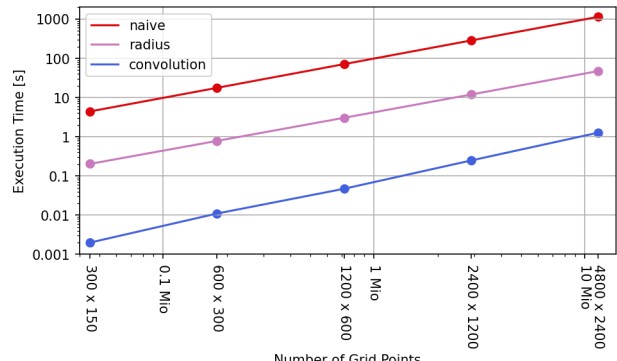

**Figure 5.** Plot of execution times from Table 2 against number of grid points. Both axis use a logarithmic scale.

in 0, its computational cost increases to a multiple of the time needed for shorter distances. With growing $\sigma$, this computation overhead is less and less noticeable in the test series.

| Gaussian Width $\sigma$ | Algorithm | | |
|---|---|---|---|
| | Naive | Radius | Convol. |
| 0.25° | 652.025 | 2.376 | 0.247 |
| 0.5° | 560.094 | 4.579 | 0.246 |
| 1.0° | 280.477 | 11.864 | 0.245 |
| 2.0° | 126.241 | 37.244 | 0.244 |
| 4.0° | 125.848 | 122.956 | 0.245 |

**Table 3.** Execution times (in $s$) of the investigated algorithms for varying Gaussian width parameters $\sigma$. The number of sample points $N = 3490$ and the grid size of $2400 \times 1200$ points with a resolution of 32 points/$^\circ$ are kept constant. The convolution algorithm applied a 4-fold convolution.

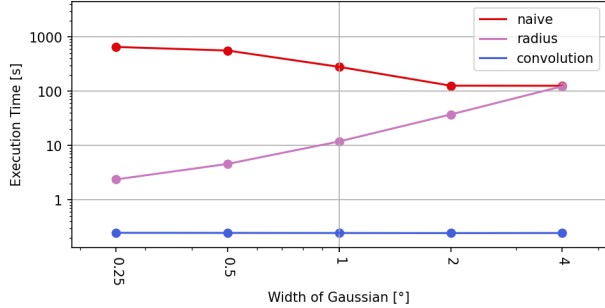

**Figure 6.** Plot of execution times from Table 3 against Gaussian width parameter. Both axis use a logarithmic scale.

## 5.4 Algorithm Fine Tuning

In the discretization step, we round the window width parameter $T \in \mathbb{N}_0$ to the nearest integer, i.e. we set

$$T = \left\lfloor \sqrt{\frac{3}{n}} \frac{\sigma}{\Delta s} + \frac{1}{2} \right\rfloor \tag{11 recap}$$

and then repeatedly convolve the input fields with a rectangular pulse signal $r_T[k]$ that has a length $2T+1$. In the extreme, if $n$ exceeds $12\sigma^2/\Delta s^2$ and hence $T = 0$, the underlying pulse signal degrades to $r_0[k]$, which is identical to the discrete unit pulse, the neutral element with respect to convolution. Under these circumstances, the algorithm stops to render a meaningful interpolation. Since normally $\sigma > \Delta s$, the respective bound for $n$ can go into the hundreds, which is why the described extreme case is typically not encountered for real applications.

Nevertheless, a general consequence of the rounding operation is that the effectively obtained Gaussian width $\sigma_{\text{eff}}$ for our algorithm only approximates the desired width $\sigma$. Let for the following considerations $u_T[k]$ be the uniform probability distribution that corresponds to $r_T[k]$, i.e.

$$u_T[k] = \frac{1}{2T+1} r_T[k].$$

By algorithmic design, $\sigma_{\text{eff}}^2$ is the variance of $u_T[k]$, which is $n$-times convolved with itself and which is employed on a grid with point spacing $\Delta s$. Therefore is

$$\sigma_{\text{eff}}^2 = n \operatorname{Var}(u_T) = n \sum_{k=-T}^{T} \frac{1}{2T+1} (k\Delta s)^2$$

$$= \frac{2n\Delta s^2}{2T+1} \sum_{k=1}^{T} k^2 = \frac{n}{3} T(T+1)\Delta s^2. \tag{12}$$

For $T \geq 1$, the integer number $T$ can be fixed by (11) to the unit interval

$$\sqrt{\frac{3}{n}} \frac{\sigma}{\Delta s} - \frac{1}{2} < T \leq \sqrt{\frac{3}{n}} \frac{\sigma}{\Delta s} + \frac{1}{2},$$

which in turn allows, when substituted into (12), to derive sharp bounds for $\sigma_{\text{eff}}$

$$\sigma^2 - \frac{n}{12} \Delta s^2 < \sigma_{\text{eff}}^2 \leq \sigma^2 + 2\sqrt{\frac{n}{3}} \sigma \Delta s + \frac{n}{4} \Delta s^2.$$

The last relation reveals that the range of possible values for $\sigma_{\text{eff}}$ grows with increasing $n$ until $\sigma_{\text{eff}}$ collapses to $0$ when $n > 12\sigma^2/\Delta s^2$, as we have seen further above.

In summary, there are two diametrically opposed effects that determine the result quality of the presented algorithm. From the perspective of the central limit theorem, the approximation performance is better for large $n$, while on the other hand $\sigma_{\text{eff}}$ tends to be closer to the target $\sigma$ for small $n$.

If full accuracy is required for the Gaussian width and the grid spacing $\Delta s$ is not strictly given in advance, one can tune the spacing such that the resulting width $\sigma_{\mathrm{eff}}$ corresponds exactly to the desired $\sigma$. For $T \geq 1$, equation (12) allows then to determine the necessary grid step as

$$\Delta s = \sqrt{\frac{3}{n\,T(T+1)}}\ \sigma.$$

If however the grid is fixed a priori and cannot be modified, the only option to attain the requested $\sigma$ exactly is to refrain from using a stringent rectangular pulse and to employ a slightly more complicated signal (Gwosdek et al., 2011). For this purpose, we set this time

$$T = \left\lfloor \frac{1}{2}\left(\sqrt{1 + \frac{12}{n}\frac{\sigma^2}{\Delta s^2}} - 1\right)\right\rfloor,$$

which is gained from taking the positive solution of the quadratic equation (12) for $T$. Now we have $n\,\mathrm{Var}(u_T) \leq \sigma^2 < n\,\mathrm{Var}(u_{T+1})$ and we define the linearly blended signal

$$
\begin{aligned}
r_{T,\alpha}[k] &= (1-\alpha)\,r_T[k] + \alpha\,r_{T+1}[k] \\
&= \begin{cases} 1 & \text{for } |k| \leq T \\ \alpha & \text{for } |k| = T+1 \qquad k \in \mathbb{Z}, \\ 0 & \text{otherwise} \end{cases}
\end{aligned}
\tag{13}
$$

for $0 \leq \alpha < 1$. This modified signal $r_{T,\alpha}[k]$ is basically the pure rectangular signal $r_T[k]$ of unit elements with a trailing element $\alpha$ appended at both ends. Due to the continuity of the variance, there must be a specific $\tilde{\alpha}$ for which $n\,\mathrm{Var}(u_{T,\tilde{\alpha}}) = \sigma^2$, whereas $u_{T,\alpha}[k] = \frac{1}{2(T+\alpha)+1}\,r_{T,\alpha}[k]$ designates the probability distribution of $r_{T,\alpha}[k]$. With

$$
\begin{aligned}
\mathrm{Var}(u_{T,\alpha}) &= \frac{1}{2(T+\alpha)+1}\sum_{k=-T-1}^{T+1} r_{T,\alpha}[k]\,(k\,\Delta s)^2 \\
&= \frac{\Delta s^2}{2(T+\alpha)+1}\left(2\alpha(T+1)^2 + 2\sum_{k=1}^{T}k^2\right) \\
&= \frac{\Delta s^2}{2(T+\alpha)+1}\left(2\alpha(T+1)^2 + \frac{1}{3}T(T+1)(2T+1)\right)
\end{aligned}
$$

we conclude that the wanted $\tilde{\alpha}$ is given by

$$\tilde{\alpha} = \frac{(2T+1)\left(\sigma^2 - \frac{1}{3}T(T+1)\,n\,\Delta s^2\right)}{2\left((T+1)^2\,n\,\Delta s^2 - \sigma^2\right)}.\tag{14}$$

The respective expression for $\tilde{\alpha}$ derived in (Gwosdek et al., 2011) corresponds to a special case of (14) when setting $\Delta s = 1$.

Remember now that the central limit theorem is valid for any PDF, which means that the mathematical framework presented in the previous chapters is also valid for the modified signal $r_{T,\tilde{\alpha}}[k]$. Therefore, we receive an optimized approximate Barnes interpolation by marginally adapting algorithm B.2 to compute the convolution with $r_{T,\tilde{\alpha}}[k]$ instead of $r_T[k]$. To do so, steps 2.

| $n$ | $T$ | $\sigma_{\text{eff}}$ | RMSE | $t_{\text{exec}}$ | $T$ | $\tilde{\alpha}$ | RMSE | $t_{\text{exec}}$ |
|---|---|---|---|---|---|---|---|---|
| | **Convolution with $r_T[k]$** | | | | **Convolution with $r_{T,\tilde{\alpha}}[k]$** | | | |
| 1 | 55 | 1.0013 | 0.3557 | 0.161 | 54 | 0.9260 | 0.3551 | 0.175 |
| 2 | 39 | 1.0078 | 0.1334 | 0.188 | 38 | 0.6868 | 0.1327 | 0.216 |
| 3 | 32 | 1.0155 | 0.0628 | 0.216 | 31 | 0.4922 | 0.0606 | 0.257 |
| 4 | 28 | 1.0282 | 0.0492 | 0.243 | 27 | 0.2083 | 0.0367 | 0.298 |
| 5 | 25 | 1.0286 | 0.0431 | 0.270 | 24 | 0.2799 | 0.0266 | 0.337 |
| 6 | 23 | 1.0383 | 0.0496 | 0.297 | 22 | 0.1256 | 0.0213 | 0.378 |
| 7 | 21 | 1.0260 | 0.0356 | 0.324 | 20 | 0.4372 | 0.0178 | 0.419 |
| 8 | 20 | 1.0458 | 0.0549 | 0.351 | 19 | 0.0956 | 0.0154 | 0.459 |
| 9 | 18 | 1.0010 | 0.0137 | 0.378 | 17 | 0.9804 | 0.0136 | 0.500 |
| 10 | 18 | 1.0551 | 0.0639 | 0.405 | 17 | 0.0316 | 0.0121 | 0.539 |
| 20 | 12 | 1.0078 | 0.0111 | 0.676 | 11 | 0.8922 | 0.0059 | 0.943 |
| 50 | 8 | 1.0825 | 0.0916 | 1.488 | 7 | 0.3125 | 0.0024 | 2.159 |

**Table 4.** Signal width parameter $T$ and effective Gaussian width $\sigma_{\text{eff}}$ respectively tail value $\tilde{\alpha}$ of the original and the optimized convolution algorithm as a function of the number of performed convolutions $n$, where $N = 3490$, grid size $2400 \times 1200$, resolution $32$ points/$^\circ$ and $\sigma = 1.0^\circ$. The root mean square error RMSE is computed for the sub-area displayed in Fig. 9 and with respect to the exact results of the naive algorithm. The execution times $t_{\text{exec}}$ are measured in seconds.

and 5. of it have to be rewritten to $h[k] = (w + \tilde{\alpha} \cdot (g[k+T+1] + g[k-T-1])) \cdot \Delta s$. Although the optimized approach requires $2L$ additions and $L$ multiplications more than the original one, the adapted algorithm 2 remains in the complexity class $\mathcal{O}(L)$. Measurements (refer to Table 4 and Fig. 7) show that the optimized interpolation in fact needs only about 10% to 30% more time for the depicted range of convolution rounds than the original one.

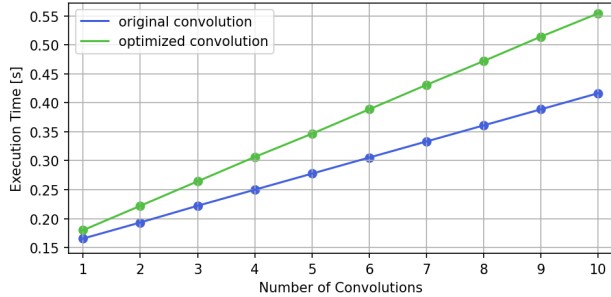

**Figure 7.** Plot of execution times from Table 4 against number of performed convolutions for both, the original and the optimized convolution algorithm.

The unstable behavior of the original convolution algorithm with respect to the number of performed convolutions can be best observed in the RMSE plot of Fig. 8, where the baseline is given by the exact interpolation of the naive algorithm. A corresponding unsteady feature is also visible in the upper row of the maps shown in Fig. 9a, more precisely in the fluctuating diameter of the small high pressure area west of the Balearic Islands.

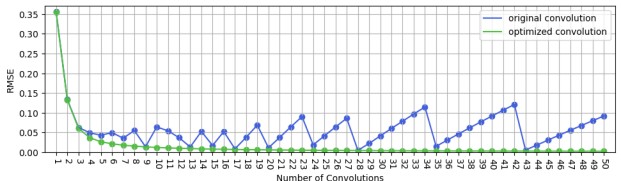

**Figure 8.** Root mean square error with respect to the exact Barnes interpolation in dependency of the number of performed convolutions for both, the original and the optimized convolution algorithm.

In contrast to that, the optimal convolution algorithm shows a stable convergence towards the exact interpolation obtained by the naive algorithm, which manifests in strictly monotonic decreasing RMSE values. These results suggest that three or four performed convolution rounds achieve already a very good approximation of Barnes interpolation when used to visualize data, as done in this paper. For other applications, which require a higher precision, a tenfold convolution or even more might make sense.

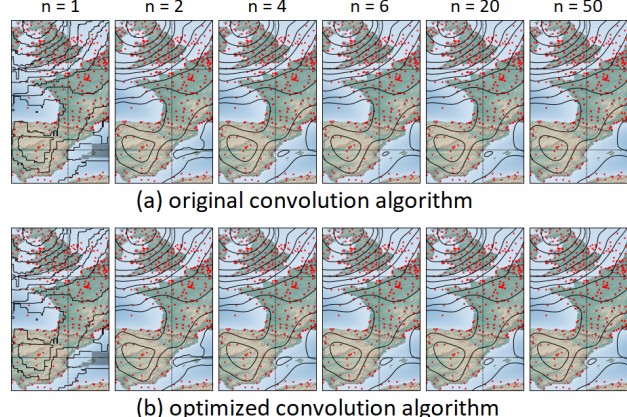

**Figure 9.** Results for different number of performed convolutions $n$. Upper row (a) with results for original convolution algorithm and lower row (b) with results for optimized convolution algorithm.

## 5.5 Application on Sphere Geometry

So far we applied the convolution algorithm on sample points contained in the plane $\mathbb{R}^2$ and using the Euclidean distance measure. For geospatial applications this simplification is acceptable as long as the considered area is sufficiently small enough.

If we deal with a dataset, which is distributed over a larger region – as it is actually the case in our test setup – it becomes necessary to take the curvature of the earth into account.

The adaptation of the naive Barnes interpolation algorithm to the spherical geometry on $\mathcal{S}^2$ consists merely in the exchange of the Euclidean distance with the spherical distance. Since the calculation of the spherical distance between two points involves several trigonometric function calls, the price of such a switchover is accordingly high and consequently exact Barnes interpolation for a spherical geometry is in our tests roughly a factor of 2.5 times slower as that with the Euclidean approach. In other words, an already costly algorithm becomes even more expensive.

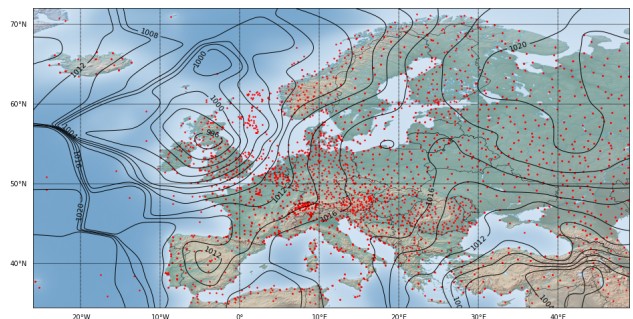

**Figure 10.** Exact Barnes interpolation with naive algorithm for the same setup as for Fig. 3 but using spherical distances from $\mathcal{S}^2$ instead of Euclidean distances from $\mathbb{R}^2$. The generated isolines are notably different in the northern part, while they are quite similar in the south.

However, the distance calculation does not occur explicitly in the convolution algorithm, since the latter by virtue of the central limit theorem is inherently tied to the Euclidean distance measure. Therefore, the convolution algorithm has to be transferred to the spherical geometry with a different method.

     The chosen approach is to first map the sample points to a map projection, which distorts the region of interest as minimal as possible and then to apply the convolution algorithm directly in that map space. The resulting field is finally mapped back 340 to original map projection and provides there an approximation of Barnes interpolation with respect to the spherical geometry of $\mathcal{S}^2$.

     Projection types that are considered suitable for this purpose are conformal map projections. Conformal maps preserve angles and shapes locally, while distances and areas underlie a certain distortion. Often used conformal maps are (Snyder, 1987)

• Mercator projection for regions of interest that stretch in east-west direction around the equator,

     • transverse Mercator projection for regions with north-south orientation,

     • Lambert conformal conic projection for regions in the mid-latitudes that stretch in east-west direction or

     • polar stereographic projection for polar regions.

In order to replicate Fig. 10 with our fast optimized convolution algorithm, we therefore use for our test setup with sample
points in the mid-latitudes a Lambert conformal conic map projection. We choose the two standard parallels that define the
exact projection at latitudes of $42.5°N$ and $65.5°N$, such that our region of interest is evenly captured by them. By nature of
Lambert conformal conic maps, the chosen map scale is exactly adopted along these two latitudes, while it is too small between
them and too large beyond them. Similarly, for a grid with a formal resolution of 32 grid points$/°$ that is embedded into this
map, the specified resolution applies only exactly along the standard parallels, while the effective resolution between them is
smaller and beyond them larger.

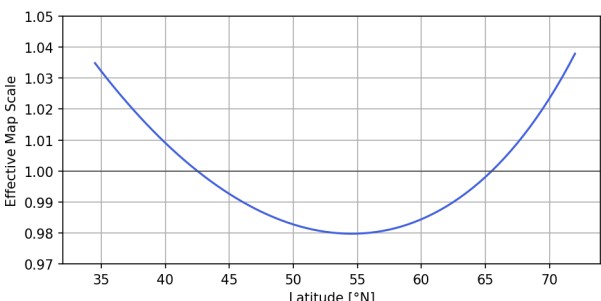

**Figure 11.** The effective scale of a Lambert conformal conic map in dependency of the latitude, if the scale at $42.5°N$ and $65.5°N$ is set to
1.0. The minimum scale of 0.98 is reached at a latitude of $54.5°N$.

We now employ the optimized convolution algorithm with a nominal Gaussian width parameter $\sigma = 1.0°$ on the Lambert
conformal conic map grid postulated above, in which we injected the given sample points beforehand. The resulting field
shown in Fig. 12 thereby experiences a twofold approximation, the first one caused by the distortion of the map and the second
one due to the approximation property of the convolution algorithm.

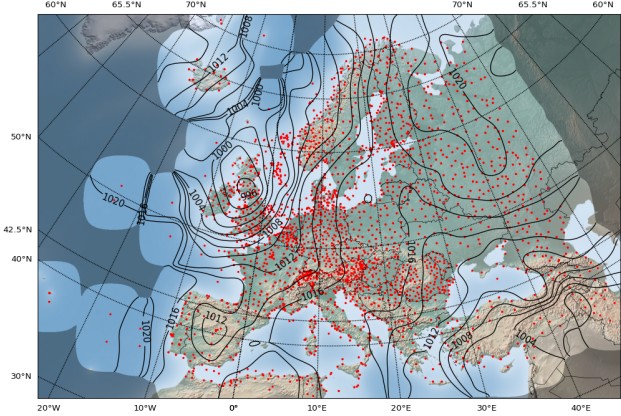

**Figure 12.** Optimized Barnes interpolation algorithm applied on Lambert conformal map on a $2048 \times 1408$ grid with a resolution 32 grid
points$/°$ along the standard parallels at $42.5°N$ and $65.5°N$, $\sigma = 1.0°$ and a 4-fold convolution.

| Target Map Grid Size | Resol. | $\mathcal{S}^2$ Algorithm | | Split Times | | Lambert Map Grid Size |
|---|---|---|---|---|---|---|
| | | Naive | Convol. | Actual Convol. | Back Proj. | |
| $300 \times 150$ | 4 pt/$^\circ$ | 10.792 | 0.005 | 0.004 | 0.001 | $256 \times 176$ |
| $600 \times 300$ | 8 pt/$^\circ$ | 42.987 | 0.020 | 0.016 | 0.004 | $512 \times 352$ |
| $1200 \times 600$ | 16 pt/$^\circ$ | 174.081 | 0.089 | 0.070 | 0.019 | $1024 \times 704$ |
| $2400 \times 1200$ | 32 pt/$^\circ$ | 700.089 | 0.408 | 0.326 | 0.082 | $2048 \times 1408$ |
| $4800 \times 2400$ | 64 pt/$^\circ$ | 2802.574 | 1.828 | 1.493 | 0.335 | $4096 \times 2816$ |

**Table 5.** Execution times (in $s$) for the naive and the optimized convolution algorithms using spherical distances on $\mathcal{S}^2$ for varying grid sizes. The number of sample points $N = 3490$ and the Gaussian width $\sigma = 1.0^\circ$ are kept constant. The convolution algorithm applied a 4-fold convolution and was executed on a Lambert map grid of the indicated size. The two split time columns show the separated execution times for the actual convolution and the subsequent back projection to the plate carrée map. For the investigated scenarios, the optimized convolution algorithm is more than 1000 times faster than the naive one.

In a last step, the result field is mapped back to the target map, which uses in our case a plate carrée projection. In order to do this, the location of a target field element is looked up in the Lambert map source field and subsequently its element value is determined by bilinear interpolation of the four neighbouring source field element values. This last operation performs an averaging and thus adds a further small error to the final result shown in Fig. 13. Similar to the case for the Euclidean distance, the comparison with the exact Barnes interpolation on $\mathcal{S}^2$ in Fig. 10 yields a very good correspondence. This perception is also

supported by measurement of the RMSE, which adds up for the same sub-area as investigated in Table 4 to 0.0467, which is negligible larger than the corresponding 0.0367 measured for the Euclidean case.

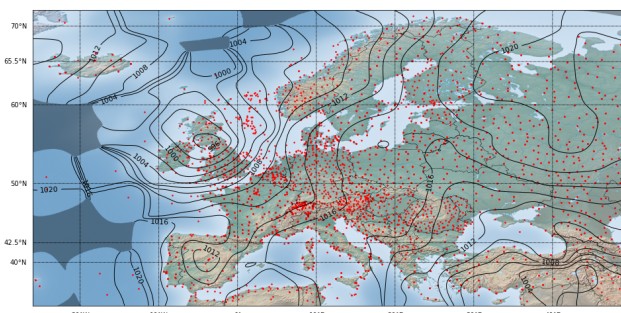

**Figure 13.** Resulting field from Fig. 12 projected back to a plate carrée map.

## 5.6 Round-off Error Issues

Computing the convolution of a rectangular pulse in floating point arithmetic using the moving window technique as described in algorithm B.2 of chapter 4 is extremely efficient, but is also prone to imprecisions since round-off errors are steadily accumulated during the progress. Different approaches are known to reduce this error. The Kahan summation algorithm (Kahan, 1965), for instance, implements an error compensation scheme, at the expense of requiring essentially more basic operations than used for ordinary addition.

Another error reduction technique that is effective in the context of Barnes interpolation, is to center the numbers to be added around $0.0$, where the mesh density of representable floating point numbers is highest. For this purpose, an offset

$$\bar{f} = \frac{1}{2} \left( \min_{1 \leq k \leq N} f_k + \max_{1 \leq k \leq N} f_k \right)$$

is initially subtracted from the sample values $f_k$, such that their range is exactly located around 0.0. The presented convolution algorithm is then applied to the shifted sample values. In a final step, the elements of the resulting field $F[i,j]$ are shifted back to the original range by adding $\bar{f}$ to each of them. This modification needs basically $N + W \cdot H$ extra additions, such that the computational complexity of the convolution algorithm is not harmed and stays unchanged.

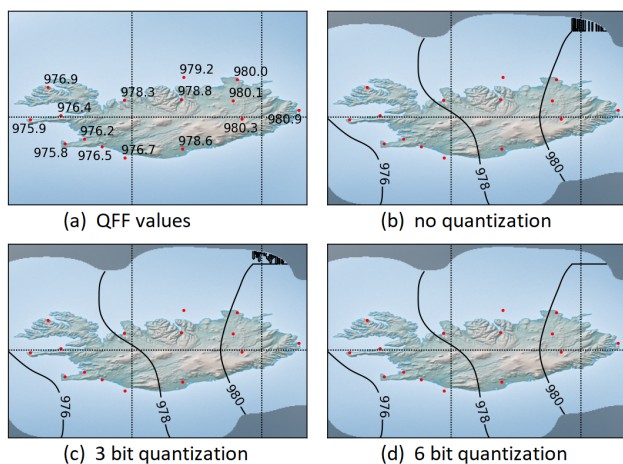

**Figure 14.** Constellation of QFF values over Iceland with generated isolines for $\sigma = 0.75$. Note the 980.0 hPa observation value in the northeast of Iceland, which triggers the creation of the faulty isoline for the same value. By increasing the quantization degree of the resulting field values, the visual appearance is gradually improved.

Minimal numerical round-off errors can generate surprisingly prominent artefacts when areas of constant or near-constant data are rendered with an isoline visualization. In such cases, the value obtained by the convolution algorithm fluctuates seemingly randomly around the true constant value, and if it happens that one of the rendered isolines represents exactly this value, the visualized result may appear like a fractal curve as shown in Fig. 14. A counter measure against this effect is to apply a quantization scheme to the resulting values, which basically suppresses a suitable number of least significant bits and rounds

the value to the nearest eligible floating point number. After this operation the obtained values in areas of constant data are actually constant and thus result in a more pleasant isoline visualization. In areas with varying data values, the quantization of the result data has no harmful impact.

## 6    Summary and Outlook

We presented a new technique for computing very good approximations of Barnes interpolation, which offers speed-up factors
for realistic scenarios that can reach into the hundreds or even into the thousands when applied on a spherical geometry. The underlying convolutional algorithm exhibits a computational complexity of $\mathcal{O}(N + W \cdot H)$, in which the number of sample points $N$ is decoupled from the grid size $W \times H$. This is a major improvement over the computational complexity of the naive algorithm of $\mathcal{O}(N \cdot W \cdot H)$. Our tests suggest that four to six iteration rounds of convolutions lead to approximations of Barnes interpolation that achieve highly satisfactory results for most applications.

    The usage of the algorithm is not restricted to $\mathbb{R}^2$ or $\mathcal{S}^2$ and it can be easily extended to higher dimensional spaces $\mathbb{R}^n$. The algorithm allows to incorporate quality measures that assign each sample point $\boldsymbol{x_k}$ a weight of certainty $c_k$, which specifies how much this point shall contribute to the overall result. To achieve this, the terms in the two sums in (1) are simply multiplied by the additional factor $c_k$, and likewise the same factors then also appear in approximation 3.
Barnes interpolation is often used in the context of successive correction methods (Cressman, 1959; Bratseth, 1986) with or without a first guess from a background field. In this technique, the interpolation is not performed just once, but applied several times with decreasing Gaussian width parameters to the residual errors in order to minimize them successively. Needless to say, that instead of exact Barnes interpolation, the convolutional algorithm can equally be used for the method of successive correction.
Since the presented solution for spherical geometries is only suitable for the treatment of local maps, we plan in a next step to generalize the approach to global maps. This could for instance be done by smoothly merging local Barnes interpolation approximation patches into a global approximation.

    Furthermore, we also want to provide a statement about the quality of the calculated approximation depending on the number of performed convolutions by deriving a theoretical upper bound for the maximum possible error. It will also be of interest,
in a similar way to Getreuer, to consider other distributions for the approximation and investigate their behavior in terms of computational speed and accuracy.

*Code and data availability.* The formal algorithms introduced in this paper are provided as Python implementation on GitHub https://github.com/MeteoSwiss/fast-barnes-py under the terms of the BSD 3-clause license and are archived on Zenodo https://doi.org/10.5281/zenodo.7651530. There are also the sample dataset and the scripts included, which allow to reproduce the figures and
tables presented in this paper. The interpolation code is as well available on https://PyPI.org as Python package fast-barnes-py.

## Appendix A: $n$-fold Self-Convolution

For an integrable function $f(x) \in L^1(\mathbb{R}) = \{f : \mathbb{R} \to \mathbb{R} \mid \int_{-\infty}^{\infty} |f(t)| \, dt < \infty\}$, the $n$-fold convolution with itself $f^{*n}(x)$ is recursively defined by

$$
\begin{aligned}
f^{*(n+1)}(x) &= f * f^{*n}(x) \\
&= \int_{-\infty}^{\infty} f(t) \cdot f^{*n}(x - t) \, dt \quad \text{for } n = 1, 2, \cdots
\end{aligned}
\tag{A1}
$$

with $f^{*1}(x) = f(x)$. The equivalent closed form representation

$$
\begin{aligned}
f^{*(n+1)}(x) &\\
&= \int_{-\infty}^{\infty} \cdots \int_{-\infty}^{\infty} f(t_1) \cdots f(t_n) f(x - t_1 - \cdots - t_n) \, dt_1 \cdots dt_n
\end{aligned}
$$

is in most cases only of formal interest, since in practice the effective calculation of the multiple integral will lead again to the recursive definition (A1).

Analogously, in the case of an integrable function of two variables $f(x,y) \in L^1(\mathbb{R}^2) = \{f : \mathbb{R}^2 \to \mathbb{R} \mid \int_{-\infty}^{\infty} \int_{-\infty}^{\infty} |f(s,t)| \, ds \, dt < \infty\}$, the $n$-fold convolution with itself $f^{*n}(x,y)$ is recursively given by

$$
\begin{aligned}
f^{*(n+1)}(x,y) &= f * f^{*n}(x,y) \\
&= \int_{-\infty}^{\infty} \int_{-\infty}^{\infty} f(s,t) \cdot f^{*n}(x - s, y - t) \, ds \, dt
\end{aligned}
\tag{A2}
$$

for $n = 1, 2, \cdots$ and with $f^{*1}(x,y) = f(x,y)$.

## Appendix B: Separable Functions

A function of two variables $g(x,y) \in L^1(\mathbb{R}^2)$ is called separable, if there exist two functions of one variable $g_1(x)$ and $g_2(y)$, both in $L^1(\mathbb{R})$, such that the following equality holds

$$
g(x,y) = g_1(x) \cdot g_2(y).
\tag{B1}
$$

The convolution of $f(x,y)$ with a separable function can be decomposed into two unidirectional convolutions that act only along one of the coordinate axis. In order to make this clear, we define two left-associative operators $\overset{x}{*}$ and $\overset{y}{*}$ that map from $L^1(\mathbb{R}^2) \times L^1(\mathbb{R}) \to L^1(\mathbb{R}^2)$ by setting

$$
f \overset{x}{*} g_1(x,y) = \int_{-\infty}^{\infty} f(s,y) \cdot g_1(x - s) \, ds,
\tag{B2}
$$

$$
f \overset{y}{*} g_2(x,y) = \int_{-\infty}^{\infty} f(x,t) \cdot g_2(y - t) \, dt,
\tag{B3}
$$

where $\overset{x}{*}$ convolves along the x-axis and $\overset{y}{*}$ along the y-axis. With these definitions we then find

$$f * g\,(x,y) = \int\limits_{-\infty}^{\infty} \int\limits_{-\infty}^{\infty} f(s,t) \cdot g(x-s, y-t)\, ds\, dt$$

$$= \int\limits_{-\infty}^{\infty} \left( \int\limits_{-\infty}^{\infty} f(s,t) \cdot g_1(x-s)\, ds \right) g_2(y-t)\, dt$$

$$= \int\limits_{-\infty}^{\infty} \left( f \overset{x}{*} g_1(x,t) \right) \cdot g_2(y-t)\, dt = f \overset{x}{*} g_1 \overset{y}{*} g_2\,(x,y).$$

From the fact that we can change the order of integration, we infer finally

$$f * g\,(x,y) = f * \Big( g_1(x) \cdot g_2(y) \Big)$$

$$= f \overset{x}{*} g_1 \overset{y}{*} g_2\,(x,y) = f \overset{y}{*} g_2 \overset{x}{*} g_1\,(x,y). \tag{B4}$$

Hence, the two operands $g_1$ and $g_2$ commute, but note here that the unidirectional operators to their left have to be swapped with them as well. The $n$-fold convolution with a separable function decomposes thus into

$$f * g^{*n}\,(x,y) = f * \Big( g_1(x) \cdot g_2(y) \Big)^{*n}$$

$$= f \overset{x}{*} g_1 \overset{y}{*} g_2 \cdots \overset{x}{*} g_1 \overset{y}{*} g_2\,(x,y).$$

Due to the commutation law (B4), we can basically write the operands on the r.h.s. of $f$ in any order, but we prefer to group them as

$$f * g^{*n}\,(x,y) = f \overset{x}{*} g_1 \cdots \overset{x}{*} g_1 \overset{y}{*} g_2 \cdots \overset{y}{*} g_2\,(x,y)$$

$$= f \overset{x}{*} g_1^{\overset{x}{*}n} \overset{y}{*} g_2^{\overset{y}{*}n}\,(x,y). \tag{B5}$$

Because the last formula looks a bit cumbersome, we again use (B4) to join the two $n$-fold self-convoluted operands to a separable function, which then reads as

$$f * g^{*n}\,(x,y) = f * \Big( g_1^{\overset{x}{*}n}(x) \cdot g_2^{\overset{y}{*}n}(y) \Big). \tag{B6}$$

Throughout the paper we use the concise representation of (B6), but keep in mind that it is equivalent to (B5), where each convolution operand is expressed separately, and thus indicates clearly how this expression is to be calculated.

## Appendix C: Generalized Approximate Barnes Interpolation

For the derivation of approximation 3, we used a separable two-dimensional PDF that consists of two uniform one-dimensional distributions. This result can be broadened, if more general marginal distributions are employed. Let for this purpose $p_1(x)$

and $p_2(x)$ be two one-dimensional PDFs with mean 0 and variance $\frac{\sigma^2}{n}$. Now we define the separable two-dimensional PDF $p(x,y) = p_1(x) \cdot p_2(y)$, which has a mean vector $\mathbf{0}$ and a covariance matrix $\frac{\sigma^2}{n}\mathbf{I}$. Consequently, the PDF $p(x,y)$ constructed in this way satisfies the assumptions of the limit theorem (6) and thus, after performing the same conversion steps as taken in chapter 3, follows the generalized

**Approximation 4.** *For sufficiently large $n$, Barnes interpolation on the Euclidean plane $\mathbb{R}^2$ can be approximated by*

$$f(x,y) \approx \frac{\left(\sum_{k=1}^{N} f_k \cdot \delta_{\boldsymbol{x_k}}\right) * \left(p_1^{\overset{x}{*}n}(x) \cdot p_2^{\overset{y}{*}n}(y)\right)}{\left(\sum_{k=1}^{N} \delta_{\boldsymbol{x_k}}\right) * \left(p_1^{\overset{x}{*}n}(x) \cdot p_2^{\overset{y}{*}n}(y)\right)}, \tag{C1}$$

*provided that the quotient is defined.*

In practice, $p_1(x)$ and $p_2(x)$ will most often be chosen to be identical. Using the normal distribution $\varphi_{0,\sigma}(x)$ with mean value 0 and variance $\sigma^2$, i.e.

$$\varphi_{0,\sigma}(x) = \frac{1}{\sqrt{2\pi}\sigma}\, \mathrm{e}^{-\frac{x^2}{2\sigma^2}},$$

it is clear from (8) that we can formulate Barnes interpolation based on a convolutional expression where even equality holds.

**Theorem 1.** *Let $\varphi_{0,\sigma}(x)$ be the normal distribution with mean value 0 and variance $\sigma^2$. For Barnes interpolation on the Euclidean plane $\mathbb{R}^2$, then holds*

$$f(x,y) = \frac{\left(\sum_{k=1}^{N} f_k \cdot \delta_{\boldsymbol{x_k}}\right) * \left(\varphi_{0,\sigma}(x) \cdot \varphi_{0,\sigma}(y)\right)}{\left(\sum_{k=1}^{N} \delta_{\boldsymbol{x_k}}\right) * \left(\varphi_{0,\sigma}(x) \cdot \varphi_{0,\sigma}(y)\right)}. \tag{C2}$$

Note that in the case of the normal distribution, it is sufficient to apply the convolution just once.

*Competing interests.* The author declares that he has no conflict of interest.

*Acknowledgements.* The author would like to thank the topical editor Sylwester Arabas and the two anonymous reviewers for their valuable comments and helpful suggestions. All map backgrounds were made with Natural Earth, which provides free vector and raster map data at naturalearthdata.com.

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
