# Peer review of "Fast Approximate Barnes Interpolation"

_Geoscientific Model Development, 2022_

## Author Response (AR1)

We would like to thank the two anonymous referees for the time and effort they took to carefully review our manuscript and for their valuable comments and suggestions that contributed to improve its quality. Below, we give our responses (in blue) to the specific comments of the referees (in black).

**Referee 1**

It is a good paper. The idea of approximating Guassian weights using the central limit theorem (CLT) is very good. The Approximation 3 is a happy result mainly because of cancellation of the common factors in the numerator and the denominator. The algorithms for implementation are also nice and thorough. It is surprising to read how the computing time is reduced from a naive Barnes interpolation. Nevertheless, I have some comments and questions, basically for a better presentation.

1. It uses a special kind of the uniform random variable on an interval with a threshold. Hope the author describe why this threshold (sqrt(3/n) sigma) was chosen.

The calculation, why the interval has the length (sqrt(3/n) sigma) is actually not obvious. It is basically chosen in this way, such that the variance of the resulting uniform distribution $u_n(x)$ is (sigma^2/n).

We added on page 3, line 65 of the manuscript a short comment about that and a calculation that verifies this.

2. page 3, line 54: hope the author provide the definition of the n-fold convolution of p(x) with itself.

We provide the definition of the n-fold convolution of p(x) with itself in the new Appendix A1 of the manuscript.

3. p5 line 92: hope the author provide more concretely what *^nx and *^yn are, because some readers may know it after reading the Algorithms 2, 3, and 4.

We fully agree, the explanation of the operators *^xn and *^yn in the text is minimalistic. We give now a thorough derivation of them in the new Appendix A2 of the manuscript

4. In Section 5, the author applied 4-fold convolution. It is known that a summation of uniform random variables converge slowly to the CLT compared to other uni-modal random variables, even though the convergence in distribution and the convergence in pdf are different. Thus I think only 4-fold is too small. I would recommend to apply at least 10-fold convolutions in real applications to ensure a good approximation.

Maybe one has to differentiate from application to application. When Barnes interpolation is used to visualize data by means of isolines - as done in the manuscript - we noticed that the resulting isolines found more or less their stable form already after applying a 3-fold convolution. For other applications, which require a higher precision, a 10-fold convolution or even more might make sense.

We extended the corresponding remark on p. 18 line 295 accordingly.

5. The result in this paper is only a nice approximation to Barnes interpolation. I think the title can be changed to 'Fast approximation to Barnes interpolation'.

We did not use the term "approximate" in the title, because other _feasible_ methods to compute Barnes interpolation are approximations as well, but do not emphasize this in an obvious way.

The title of the manuscript is now "Fast Approximate Barnes Interpolation".

6. It uses a special kind of the uniform random variable and a numerical integration for convolution in Algorithm 2. It is still good, but  I guess one may consider other random variables which may accelerate the convergene to the CLT. Then numerical integration may be a little more complex than that in this paper. This may increase computing time in Algorithm 2, but may be alright with a smaller n.

In fact, there are many other PDFs that have a faster CLT-convergence to a Gaussian than the uniform PDF - in the extreme case one could even take a normal distribution itself for which we would have n = 1.

But when using a non-trivial PDF, it is clear that putting and moving the weight window (as described on p. 9 line 161) over the data to be convolved requires in general 2T+1 multiplications and 2T additions per element. Thus, the simplicity of algorithm 2 is lost to a far part and the algorithmic complexity of it grows to O(L*T) instead of O(L). Overall we then could not claim a complexity of O(N+W*H) anymore, this would be rather O(N+W*H*T).

Your comment leads us to consider to add a remark on p. 6 line 128 that Approximation 3 is valid in a much more general context, i.e. also for other (non-uniform) PDFs. In the case of a normal distribution even equality holds. These relationships are now described in the new Appendix A3 of the manuscript.

7. In Algorithms 2 and 3, I was a bit confused with the notations g, r_T, and F[i,j]. Hope the author kindly give short descriptions on these notations to improve reader's understanding.

These notations basically refer to the formulae and text given around the Algorithms.

We addressed this problem in chapter 4 Discretization by first adding a hint about the notation that we use round parentheses for functions g(x) defined on the continuum and square brackets for discrete functions g[x]. Throughout the chapter, we carefully improved the readability by giving more context here and there and by being more specific.

8. Finally, I think that the idea of approximating Guassian weights using the CLT is very good and so it can be applied to the other area of numerical computations which use Gaussian weights. Hope the author try to search the literature if any body has already published or applied this idea.

This is a good question. The CLT is used by some applications to quickly generate the weights of a normal distribution (as above, of course only in an approximate way). Further we assume that some image processing programs like Photoshop employ this technique in order to apply a Gaussian blur filter on an image. This process can be regarded as a special case of the method presented in this manuscript, where the observation points in

question are given by the image matrix. In this case the points are located on a regular grid and the denominator of equation (8) simplifies to 1.

We refer in the last paragraph of the Introduction to applications of the CLT in image processing and computer vision (Wells, 1986). Further we refer in chapter 5.4 on p. 16, line 269 and on p. 17, line 283 to (Gwosdek et al., 2011), where in principle the same considerations are made for so called "extended box filters". Finally, we mention in the Conclusions (Getreuer, 2013), who investigated different "Gaussian convolution algorithms".

The References were updated with (Getreuer, 2013), (Gwosdek et al., 2011) and (Wells, 1986).

**Referee 2**

This is a nice paper. It presents an efficient method to perform a Gaussian smoothing, which takes advantage of the CLT by using the repeated convolution of a computationally cheap (uniform) smoothing to approximate the desired Gaussian result (and further that the 2D result is the product of two 1D calculations).

I see that the author already improved the first version of the manuscript in response to previous comments, and I found it very readable. I found that the questions that arose on my first reading were answered later on. I have a minor suggestion about the presentation of the computational complexity of the main result, which is that the wording could be more precise. For example, in the abstract "When implemented naively, the computational complexity of Barnes interpolation depends directly on both the number of sample points and the number of grid points." This initially confused me as all algorithms (including the proposed one as well as the naive approach) must surely depend directly on these numbers, simply from the time taken to read and write them. It would be better expressed as varying with the \*product\* of the number of sample points and grid points. Or else (additionally?) write it in O() notation so there can be no confusion. The Abstract should also explicitly present the complexity of the new result (and perhaps both old and new complexities could be placed again in the Conclusions).

We agree that this can be written more precise and also should be written with more emphasis, as it is ultimately the main achievement of the presented manuscript. Hence, the Abstract now contains the computational complexity in explicit O() notation for both, the naive approach and the newly presented method.

As you suggested, the first paragraph of the Conclusions now also contrasts the computational complexity of the naive approach and the presented approach.

Also one piece of ungrammatical English at l133: "are sufficiently good contained" could be "sufficiently well contained" perhaps?

This is corrected, thank you for the hint.

I suggest that the Conclusion contain a repetition of the author's suggestion that 4 convolutions seems to work well in most practical applications.

We agree with this. This suggestion is now also included in the first paragraph of the Conclusions.

---

## Author Response (AR2)

**Response to Comments from Sylwester Arabas**

The author would like to thank the topical editor Sylwester Arabas for his valuable comments and helpful suggestions that contributed to improve the quality of the manuscript. Below, you find the responses (in blue) to the specific comments of the topical editor (in black).

**Comments from Sylwester Arabas**

Thank you for addressing the reviewers' comments in the revised manuscript and for providing the point-by-point reply. Let me also underline that I am sorry that the review process took longer than expected. Congratulations for excellent reviewers' scoring: in six out of eight received marks the paper was ranked as "Excellent".

Below, I'm listing technical remarks I'm hereby asking to address before moving on with acceptance, typesetting and publication:

1. Please attach a version number for the software release, e.g. v1.0 and ensure that the same version number and license metadata are featured on Zenodo. As of now, the GitHub repository has a release without an explicit number, while the only tag indicates a pre-release: v0.8-alpha.

There is now a new release on GitHub named fast-barnes-py v1.0.0 with a corresponding tag v1.0.0. The corresponding archive on Zenodo uses the same name.

2. While it is in not required for publication, I highly recommend to consider disseminating the software as a Python package (by including a setup.py, __init__.py, pyproject.toml files). This way, users could easily import the code with proper version identification. This would also enable dissemination of the code on PyPI.org or conda forge package sites.

The interpolation code is available on PyPI.org as Python package fast-barnes-py. The current version number is 1.0.0.
This is now also mentioned in the "Code and data availability" section. For some unknown reason the latexdiff program did not highlight this difference – just to let you know.

3. According to the GMD guidelines (https://www.geoscientific-model-development.net/about/manuscript_types.html#item2), the following applies to papers submitted as "Development and technical paper": "If the main intention of an article is to make a general (i.e. model independent) statement about the usefulness of a new development, but the usefulness is shown with the help of one specific model, the model name and version number must be stated in the title." Please thus append to the title a phrase or a subtitle featuring the version number, e.g.: "Fast approximate Barnes interpolation and its Python/Numba implementation fast-barnes-py v1.0" (or alike).

The title is now "Fast Approximate Barnes Interpolation: Illustrated by Python/Numba Implementation fast-barnes-py v1.0".

4. Language/typesetting remarks (disclaimer: I'm not a native speaker; note also that as any other accepted GMD manuscript, the paper will go through English copy-editing before galley proofs):

Corrected as suggested, unless commented otherwise below.

- p1/l15: change "commonly" to "jointly"
- p4/l85: rephrase around "writes" (?), e.g. with "can be expressed as" or "reduces to"
Rephrased with "becomes".
- p5/l88: remove "better"
- p6/l97: comma after "definition"
- p7/l140: comma after "distribution", rephrase around "even equality holds"
Rephrased to "In the special case of using a normal distribution with mean 0 and variance \sigma^2, we can even formulate a convolutional expression that is equal to Barnes interpolation. Refer to appendix C for more details.
- p9/l191: comma before "the outer loop"
- p10/l193: change "After," into "Subsequently, "
- p10/l201: write "NaN" without italics
- p11/l207: change "in the further discourse" to "hereinafter"
- p11/l207: change "in an experimental setup" to "on a dataset" (otherwise unclear if experimental refers to testing or measurements)
- p12/l220: change "reduce the unsteadiness of purely interpreted" to "achieve compiled-code performance using ordinary"
- p12/l225: change "Barnes' " into "Barnes " (as used elsewhere)?
- p13/l236: comma after "In this sense"
- p13/l247: comma after "For smaller grid sizes"
- p16/l283: change "Summarized can be stated that" to "In summary, " or "In summary, there are ... that determine"
- p17/l309: comma after "Therefore"
- p17/l317: change "This finds also its visual correspondence" to "A corresponding feature is visible" (?)
- p18/Table 4: rephrase "Key numbers"
- p18/Table 4: change "in dependency" to "as a function of" or alike
Changed figure caption of Table 4 to "Signal width parameter T and effective Gaussian width \sigma_eff respective tail value \tilde{\alpha} of the original and the optimized convolution algorithm as a function of the number of performed convolutions n, where N = 3490, grid size 2400 x 1200, resolution = 32 points/° and \sigma = 1.0°.
- p19/l327: change "data, which is" into "data, which are" or "dataset which is"
- p19/l333: change "effortful" into "costly" or alike
Rephrased to "…an already costly algorithm becomes even more expensive."
- p22/l371: remove "intelligent"
- p22/l373: rephrase to "Another error reduction technique which is effective..."
- p23/l389: change "which promise" into "which offers"
- p24/l414: change "earlier" to "in the paper"
- p24/l109: move "Getreuer" out of the parentheses

5. Reading through the final sentences of the conclusions, I found it a bit puzzling to discern which sentences refer to the paper and which refer to outlined future endeavors. Perhaps naming the section "Summary and outlook", and separating the summary from the outlook remarks in a clearer manner would improve flow?

This can in fact be better structured. Renamed the paragraph as proposed to "Summary and Outlook" and inserted an empty line between the two parts.

6. Algorithms: please add a number for the algorithm on page 2 and use it when referring to it (all other algorithms are numbered).

The naive algorithm is now labeled with "Algorithm A" and the convolutional algorithm with "Algorithm B", whereas its sub-procedures are named "Algorithm B.1, B.2 and B.3". In this way it is clear for the reader that the B-algorithms build together a program unit.

7. Figures: please use vector graphics format instead of raster graphics (e.g., by saving to pdf, postscript or svg formats).

I agree, vector graphic formats are resolution independent. Unfortunately most figures are were created by Python's matplotlib and are therefore raster images. However, the used figures are stored in lossless .png format and have a resolution of 300dpi, if printed with a width that corresponds to the column width of GMD papers.

8. "Python" remarks:

- p15/l259 (computational cost increases in Python): if the code is compiled by Numba/LLVM, this would not be a Python behavior anymore - please clarify.

Good remark. I noticed this effect in fact also in my Java implementation and thus it seems to be a general behavior. I dropped therefore the reference to Python programming language.

- p10/l195 & p12/l221 (potential for concurrent computations): just a comment for future developments (outside of the scope of the paper) - it seems likely that Numba `prange` constructs could facilitate introducing concurrency in the code (see https://numba.pydata.org/numba-doc/latest/user/performance-tips.html#parallel-true) by just changing selected `range` statements into `numba.prange` (akin to OpenMP directives).

I agree, this is subject for future work.

- Also, please consider either specifying or allowing users of the package to specify performance-oriented `njit` parameters such as `error_model='numpy'` (instead of the default 'python') and `fastmath=True` (assuming the code does not rely on NaN propagation, etc) - again, this is just a comment, outside of scope of the paper.

Thanks for the hints, this is subject for future work.

9. Appendices: please consider changing the nested numbering in appendices (A: A1, A2, A3) into Appendix A, Appendix B and Appendix C.

Done as proposed.

10. In the bibliography:

- for Bentley 75, please replace the url with https://doi.org/10.1145/361002.361007
- for Bergthórsson et al., please correct the DOI to:
https://doi.org/10.3402/tellusa.v7i3.8902
- for Daley, the DOI refers to a review of the book, and not the book itself

- for Gwosdek et al., please correct DOI to https://doi.org/10.1007/978-3-642-24785-9_38
- for Koppert, please consider adding an URL:
https://ams.confex.com/ams/84Annual/techprogram/paper_71789.htm
- for Wells 86, the "dx." in the URL can be removed

Thanks for double checking the links. Corrected as suggested.